# Microbial Diversity and Authigenic Mineral Formation of Modern Bottom Sediments in the Littoral Zone of Lake Issyk-Kul, Kyrgyz Republic (Central Asia)

**DOI:** 10.3390/biology12050642

**Published:** 2023-04-23

**Authors:** Anton Maltsev, Darya Zelenina, Alexey Safonov

**Affiliations:** 1V.S. Sobolev Institute of Geology and Mineralogy Siberian Branch, RAS, Novosibirsk 630090, Russia; 2A.N. Frumkin Institute of Physical Chemistry and Electrochemistry, Russian Academy of Sciences, Moscow 119071, Russia

**Keywords:** authigenic minerals, Salt Lake, Issyk-Kul, microbial biodiversity, biogeochemistry, bottom sediments, early diagenesis

## Abstract

**Simple Summary:**

A comprehensive geochemical, mineralogical and microbiological analysis of bottom sediments was carried out in five main areas of the littoral zone of Lake Issyk-Kul, one of the largest mountain lakes in the world. The aim of this work is to study the microbial diversity of bottom sediments of Lake Issyk-Kul and to establish the role of microbiota in biogeochemical processes, in particular, its involvement in authigenic mineral formation. Aerobic and anaerobic oxidation of organic matter begins at the water-sediment interface and is the main source of carbon dioxide which provides the formation of carbonate minerals. Oxygen consumption during organic carbon oxidation triggers anaerobic sulphate reduction processes, leading to the formation of authigenic sulphide minerals. This work is the initial stage in the study of the microbiology and biogeochemistry of the ecosystem of Lake Issyk-Kul, and the data obtained will allow a better understanding of the processes of C, N and S cycling in the lake and the mechanisms of formation of modern bottom sediments.

**Abstract:**

This article presents geochemical, mineralogical and microbiological characteristics of five samples of modern bottom sediments in the littoral zone of the high-mountain salty lake Issyk-Kul. The 16S rRNA gene sequencing method shows that the microbial community consists of organic carbon degraders (representatives of phyla: *Proteobacteria*, *Chloroflexi*, *Bacteroidota* and *Verrucomicrobiota* and families *Anaerolineaceae* and *Hungateiclostridiaceae*), photosynthetic microorganisms (representatives of *Chloroflexi*, phototrophic *Acidobacteria*, purple sulphur bacteria *Chromatiaceae* and cyanobacteria) and bacteria of the reducing branches of the sulphur biogeochemical cycle (representatives of *Desulfobacterota*, *Desulfosarcinaceae* and *Desulfocapsaceae*). The participation of microorganisms in processes in the formation of a number of authigenic minerals (calcite, framboidal pyrite, barite and amorphous Si) is established. The high diversity of microbial communities indicates the presence of labile organic components involved in modern biogeochemical processes in sediments. The active destruction of organic matter begins at the water-sediment interface.

## 1. Introduction

The problem of studying natural processes of mineral formation and the geochemical concentration of chemical elements associated with the activity of microorganisms is widely discussed in the world literature on geological microbiology, biogeochemistry and biomineralogy [1,2,3,4,5]. Of particular interest is the study of the microbiological characteristics of saline lakes of recreational importance. A comprehensive study of the biogeochemical properties of the bottom sediments of these lakes is one of the current areas of modern microbiology [6,7,8]. In brackish and saline lakes, high mineral content ensures a massive development of microbiota, especially in coastal zones, where the number and species composition of aquatic microorganisms increases dramatically [4]. Due to shallow depths in the littoral zone, the decomposition of organic matter (OM) in the water column is incomplete and suspended matter containing labile OM is delivered to the bottom, which is actively used by various physiological groups of microorganisms [9].

Organic matter entering bottom sediments during sedimentation undergoes a series of structural transformations and is involved in redox processes aimed at establishing a balance between the oxidised mineral component of the sediment and OM, which acts as a reducing agent [10]. Thus, the main changes in the mineral part of the bottom sediments are directly or indirectly related to organic matter and primarily occur through the energy of microbial as well as chemical processes of destruction and mineralization of organic matter in the sediments. Intense microbial processes are accompanied by a sharp decrease in oxygen and activation of sulphate-reducing and methane-forming microorganisms in the surface layer of sediments, which carry out the terminal phase of OM decomposition and participate in the formation of authigenic carbonate and sulphide minerals [11]. Furthermore, the activity of various physiological groups of microorganisms leads to changes in pH and Eh values, metamorphism of the sedimentary liquid phase and redistribution of chemical elements, which leads to deposition of other authigenic minerals-barite [12] and amorphous silica [2]. This is why the study of the composition, structure and metabolic capacities of the microbial communities inhabiting the sediments of saline lakes is key to understanding the mechanisms of biogeochemical cycles and the evolution of the biosphere.

Issyk-Kul is one of the largest and deepest mountain saline lakes in the world, an oligotrophic lake with low productivity [13]. Due to its unique geophysical and ecological characteristics, this lake is of great scientific importance. To date, there are few microbiological studies of the water and sediments of Issyk-Kul. In 2002, Braissant and colleagues [14] discussed the role of microorganisms in vaterite deposition (polymorph CaCO_3_) in Lake Issyk-Kul. In 2021, Rojas-Jimenez and colleagues [15] investigated the diversity and composition of the bacterial community along a vertical gradient. In water samples from Lake Issyk-Kul affected by oil contamination in the area of the Balykchy oil depot [16], a microbial community with predominance of oil-degrading bacteria from genera Pseudomonas, Micrococcus [17] and others has been found. The role of microbiological processes in the formation of authigenic minerals in the bottom sediments of Lake Issyk-Kul, in comparison with other saline lakes [18,19,20,21], is practically unstudied.

Studies [2,6,9,14,18,19,20,21] show that bottom sediments of saline lakes in Eurasia contain a number of authigenic minerals: calcite, pyrite, rhodochrosite, siderite, barite, etc., which are formed with the direct participation of microorganisms. This process proceeds especially intensively in the upper intervals of bottom sediments in the littoral zone. We assume that the leading role in the formation of authigenic minerals in Lake Issyk-Kul belongs to various physiological groups of microorganisms: heterotrophic, sulphate-reducing, nitrifying and others. The energy source for these processes is the decomposition of organic matter. In connection with this, the main aim of our work is:To study the biogeochemistry of modern bottom sediments in the littoral zone of Lake Issyk-Kul;Determine the phylogenetic diversity of microbial communities of bottom sediments based on 16S rRNA gene sequence analysis;Identify authigenic minerals in bottom sediments and relate the diversity of different physiological groups of microorganisms in biogeochemical processes of the littoral zone (organic matter decomposition, chemical elements distribution, pore water transformation, etc.).

## 2. Physiographic Background and Methods

### 2.1. Physiographic Background

Lake Issyk-Kul is located in the northern Tien Shan of the Kyrgyz Republic (approximately 77° E and 42°30′ N), in the heart of Central Asia. It lies at an altitude of 1607 m above sea level and is surrounded by high mountain ranges: the Kungei Alatau Range in the north with the highest peaks reaching 4770 m, and the Terskei Alatau Range in the south with peaks exceeding 5200 m [22]. The lake is a Ramsar site with globally significant biodiversity and is part of the Issyk-Kul Biosphere Reserve. Lake Issyk-Kul is 182 km long, up to 60 km wide and has an area of 6236 km^2^. Its depth reaches 668 m [23]. Some 118 rivers and streams flow into the lake; the largest of them are the Djyrgalang and the Tyup [24]. It is fed by springs, including many hot springs, and by melting snow. Currently, the lake has no outlet. The lake is saline (6 g/L), oligotrophic to ultra-oligotrophic (~2–3.8 µg/L phosphorus) and has high dissolved oxygen values (6.5–7.5 mg/L at the lake bottom) [25].

Lake Issyk-Kul is characterized by a limited development of an active sedimentation zone, which is located in a small strip along its shoreline due to its location in an intermountain depression [26]. Terrigenous material plays a major role in the lake’s sedimentation process. Bottom sediments in the deep-water zone of the lake are composed of massive light-brown silty clays and contain very few sand particles (2.8%), whereas in shallow-water sediments, by contrast, the maximum sand content reaches 93.8% [27]. In the lake itself, chemogenic and biogenic sediments are formed, the proportion of which in the total accumulation is not precisely determined [26].

The sampling for the microbiological and geochemical studies was carried out in September 2021 in different areas of the lake, 15–30 m from the shore (Figure 1).

### 2.2. Research Methods

#### 2.2.1. Sampling

The presence of the highest amount of organic matter in the bottom sediments determined the choice of sampling points. As a result, small bays covered with macrophytes and charophytes were selected in the littoral part of the lake (Figure 1). In other parts of the lake the littoral sediments are mainly represented by sands. Samples were taken in the following locations: IK-1. Issyk-Kul rayon, near Toru-Aygyr village; IK-2. Tyup district, near Ak-Bulak village; IK-3. Jety-Oguz district, Pokrovsky bay; IK-4. Jety-Oguz district, near Tosol village; and IK-5. Ton district, near Kara-Talaa village. In each location, water samples and bottom sediment samples were collected for geochemical and microbiological analysis. Lake water samples were collected from a depth of 0.5–1.5 m. Bottom sediments were sampled from the upper 5–10 cm layer. Strongly hydrated black or dark grey silt constituted the sediment. We were able to obtain sediments for point IS-1 from depths of 0–5 cm and 25–30 cm. The choice of the upper sediment layer was based on the fact that it contains the maximum number and variety of microorganisms, the number of which decreases sharply deeper into the sediment [2,4,12].

Water samples were taken with a bathometer in the near-bottom horizon. Water samples for hydrochemical analysis were not treated, while for elemental analysis water, was filtered under a vacuum through 0.45 μm filters and packed into plastic bottles with the addition of concentrated nitric acid (1 mL/L) for preservation. Bottom sediment samples were taken at the littoral using a special sampler of original design. The sampler allows sediments to be taken in a plastic tube to a depth of 0.5 m. New tubes were used for each sample to avoid or minimise cross-contamination. For sampling for microbiological analysis, clean sterile containers made of polymeric materials (polypropylene) that do not affect the vital functions of microorganisms were used. A continuous core of 30 cm length was obtained only for the sample from point IK-1.

#### 2.2.2. Analytical Methods

Temperature, pH and Eh were immediately measured in the samples using an Anion 4100 ionometer. Pore water was squeezed from 5-cm pieces of sediment into oxygen-proof sealed syringes using a standard technique [10] using a 6 cm diameter mould and an Omec PI.88.00 hydraulic press (Italy).

The anion concentration in lake and pore water samples was determined by titrimetry (HCO_3_^−^) and capillary zonec electrophoresis (SPE) (Cl^−^, SO_4_^2−^, PO_4_^3−^). Cations (K^+^, Na^+^, Ca^2+^, Mg^2+^ and Li^+^) and major and trace elements (Al, Ba, Fe, Mn, Mo, Si and Sr) were measured by inductively coupled plasma atomic emission spectroscopy (ICP-AES) on a Thermo Jarrell IRIS Advantage ICP-AES spectrometer (Thermo Jarrell Ash Corp., Franklin, MA, USA).

The total dissolved carbon (TDC) and the proportions of dissolved inorganic and organic carbon (DIC and DOC, respectively) in water were determined on an Analytik Jena AG Multi N/C 2100S (Analytik Jena GmbH, Jena, Germany). The TDC was estimated from the amount of CO_2_ released from the samples after catalytic oxidation at 950° in the presence of oxygen flow, in a quartz reactor. The DIC was estimated from the amount of CO_2_ released from the samples after digestion in 10% H_3_PO_4_. The DOC was defined as the difference between TDC and DIC.

For all sampling points (except IK-1) the chemical composition of the bottom sediments was determined in the total sample. The bottom sediments from IK-1 had a heterogeneous material composition. The upper part of the sample consisted of black silt and the lower part of grey silt (Appendix A). Therefore, the upper (0–5 cm) and lower (25–30 cm) parts of the bottom sediments were taken for chemical analysis. Pore water chemistry and microbiology were only studied in the upper 5-cm part of the sediment.

The content of biogenic elements C, H and N in the bottom sediments was measured according to the method [28] on a CHNS Euro EA 3000 automatic analyzer (EuroVector S.p.A., Milan, Italy). The total organic carbon (TOC) was determined using the Tyurin method. This is a widely used method in analytical chemistry which is based on the oxidation of organic matter by a solution of potassium dichromate in sulphuric acid and the subsequent determination of Cr (III) equivalent to the organic matter content. The determination is carried out by the titrimetric method. For more complete carbon oxidation of organic compounds by potassium dichromate solution, AgSO_4_ was used as a catalyst, which allowed us to determine 95–97% of the carbon of organic compounds. The use of this technique for the determination of TOC makes it possible to exclude the influence of carbonate carbon.

The sulphur content of sediment samples—total sulphur (Stotal), sulphate (S (VI)) and sulphide (S (II))—was studied using ICP-AES. Stotal was determined by high temperature digestion in HNO_3_ under cover and then by digestion in HCl, which converts the sulphide to sulphate. S (II) was removed from the samples by boiling them in dilute HCl followed by filtration of the residue, leaving only sulphate sulphur (S (VI)). The amount of S (II) was estimated as the difference between Stotal and S (VI).

The concentrations of major (Al, Ca, Fe, K, Mg, Mn, Na and Si) and trace elements (Ba, Mn and Sr) in the bottom sediment samples were determined by ICP-AES. Grain morphology and elemental composition were studied in selected samples by scanning electron microscopy (SEM) on a Tescan Mira 3 LMU microscope. The samples were dried and ground to powder for vacuum spraying. The Quorum sputtering machine was used. It allows the deposition of ultra-thin carbon coatings on samples by cathodic and thermal spraying under medium and high vacuum conditions to produce a conductive surface. We used the characteristic X-ray mode, which allows us to conduct X-ray microanalysis and obtain data on the elemental composition of the sample [29].

#### 2.2.3. High-Throughput Sequencing of 16S rRNA Genes

The samples for total microbial DNA analysis were obtained from bottom sediments, fixed with 96% ethanol and stored at −20 °C until DNA extraction. DNA was extracted using the FastDNA™ SPIN Kit for Soil (MP Biomedicals, Irvine, CA, USA) according to the manufacturer’s instructions. A culture of E. coli K-12 served as a control for DNA extraction and 7 ng of DNA was extracted from a biomass of 10^7^ cells.

Libraries of amplicons of the V4 region of the 16S rRNA gene were prepared using primer pair 515F (5′-GTGBCAGCMGCCGCGGTAA-3′; -Pro-mod-805R (5′-GGACTACHVGGGTWTCTAAT-3′. Amplification was performed by a real-time PCR on a CFX96 Touch instrument (Bio-Rad, Hercules, CA, USA) using qPCR mix-HS SYBR (Eurogen, Russia). Denaturation, primer annealing and chain lengthening for regions V3-V4 were performed at 96, 54 and 72 °C, respectively. The steps for region V4 were performed at 96, 58 and 72 °C, respectively. The purification of the desired product from each batch was performed using Agencourt AMPure XP magnetic particles (Beckman Coulter, Brea, CA, USA).

Libraries were sequenced on a MiSeq system (Illumina, CA, USA) using a 150 nucleotide paired-end read cartridge and reagent kit (MiSeq Kit v2, 500 cycles, Illumina, San Diego, CA, USA). Libraries were prepared for each sample and sequenced in two replicates. An amplicon sequence variant (ASV) table was constructed using the Dada2 script and the SILVA 138.1 database [30]. The ASV table was analysed using MicrobiomeAnalyst software (https://www.microbiomeanalyst.ca/ (accessed on 14 November 2022)) [31].

Diversity indices (Simpson, Shannon, Equitability) were calculated using PAST 4.06b software. OTUs derived from 16s rRNA sequencing data in the SILVAngs database (https://ngs.arb-silva.de/silvangs/ (accessed on 14 November 2022)) were used to calculate diversity indices.

#### 2.2.4. Geochemical Modelling

The program PHREEQC 2.18 (pH-REdox-EQuilibrium) [32] was used to determine the speciation of elements in solutions and saturation indices. The PHREEQC is based on solving systems of mass action law and material balance equations. Saturation indices for solid mineral phases of investigated radionuclides were estimated in simulations. The saturation index, SI, is the difference between the decimal logarithms of the current products of the ionic activity of the i-th phase and the solubility constant of the corresponding mineral: If the logarithm of SI exceeds 0.5, one can predict the formation of the given phase; if it is below −0.5, its dissolution is likely. For geochemical modeling, the thermodynamic database (TDB) llnl (Lawrence Livermore National Laboratory https://rdrr.io/cran/phreeqc/man/llnl.dat.html (accessed on 14 November 2022)) was used.

## 3. Results

### 3.1. Physico-Chemical Characteristics of Lake and Pore Waters

The surface waters of Lake Issyk-Kul belong to the chloride-sulphate class of sodium group, by redox conditions to oxidized water type (Eh from +0.195 to +0.276 V). According to the alkaline-acid conditions, the waters of Lake Issyk-Kul belong to the alkaline class (pH from 8.6 to 8.7) and according to the total mineralization (5.6–6.7 g/L), they belong to the brackish water family (Figure 2). The exception is the surface waters of Point IK-3, sampled in the vicinity of Pokrovsky Bay. These are classical bicarbonate-calcium low-mineralized waters with pH 7.5, typical for freshwater lakes of Central Asia.

The value of the total alkalinity of the lake is mainly determined by the content of HCO_3_^−^ anions (0.28–0.36 g/L) and partially by CO_3_^2−^ (0.02–0.19 g/L). Differences in ionic composition and total mineralization of lake waters from different sampling points are determined by: 1. physical-geographical and hydrological factors; and 2. groundwater entering the lake in the eastern and southern parts of the coast, which contain high concentrations of chloride-, sulphate- and sodium-ions [33].

The liquid phase (pore water) of the sediments of Lake Issyk-Kul inherits the chemical composition of lake waters (Figure 2). However, it also has a number of differences. First of all, there is the decrease of the total mineralization for points IK-1, IK-2 and IK-4 which is determined by the decrease of the share of all cations. It can be connected with a peculiarity of an absorbing complex of sediment. Points IK-3 and IK-5 are characterised by an increase in mineralisation and main cations. In most of the bottom sediments, a decrease of SO_4_^2−^ and an increase of PO_4_^3−^ values is observed. This is a consequence of bacterial reduction of sulphates and microbial decomposition of OM in the sediments. All of the bottom sediments are characterised by a pH value decrease from alkaline (8.4–8.8) to neutral (7.1–7.6), ecological conditions decrease (Eh) in the range from −150 to −352 mV. Changes in physico-chemical parameters of the environment are also a consequence of microorganism activity.

The peculiarity of Lake Issyk-Kul’s water is a very low content of dissolved organic carbon (DOC) at the time of sampling (Figure 2). In lake waters, the basis of total dissolved carbon (TDC) is dissolved inorganic carbon (DOC). Pore waters are characterized by an increase of organic forms of carbon up to 6–140 mg/L, which is a consequence of organic matter breakdown. However, the low content of OM in sediments does not allow the organic forms of carbon to predominate over its mineral forms (with the exception of point IK-1).

Macro- (Fe, Si, Sr) and microelements (Al, Ba, Mn, Mo) are the most important components of the hydrogeochemical originality of limnogeosystems. The surface waters of Lake Issyk-Kul are characterized by large variations in the contents of Fe (45–570 µg/L) and Mn (1.9–117.3 µg/L) (Figure 3). The surface waters are characterized by very high Si contents (2478–7188 μg/L). Lake waters at sampling point IK-2 have high Al contents equal to 75.2 µg/L. Pore waters are characterized by an increase in the contents of almost all chemical elements in comparison with lake water (Figure 3). The maximum increase in dissolved trace elements in pore water was established for freshwater Pokrovsky Bay (IK-3). This is a consequence of the transformation of bottom sediments: the processes of leaching, cation exchange, destruction of organic matter, etc.

### 3.2. Geochemical Characteristics of Bottom Sediments

The content of total organic carbon (TOC) in the sediments of Lake Issyk-Kul is low (0.2–1.9%) (Table 1). The highest TOC contents (1.9%) are found in the desalinated part of the lake, in the area of Pokrovsky Bay (IK-3). The bottom sediments near the shoreline of the lake are characterized by low TOC contents due to the small input of autochthonous and allochthonous OM into sediments, despite the large number of macrophytes and Chara algae. For the bottom sediments at point IK-1, a decrease in TOC content with depth was found. Higher total carbon (TC) contents in the bottom sediments of Lake Issyk-Kul are associated with carbonate content at all sampled points (Table 1).

The forms of sulphur are presented in Table 1. S (II)’s reduced compounds predominate in bottom sediments composition: sulphur in Fe sulphides, H_2_S, etc. In modern bottom sediments of Lake Issyk-Kul, the presence of reduced forms of sulphur is found up to 0.08–0.29%. This indicates that the processes of bacterial reduction of sulphates are already taking place in the uppermost intervals of the sediments.

In the bottom sediments at IK-1, an increase of reduced forms of sulphur and a decrease of oxidised forms of S are observed with depth. The proportion of reduced forms of sulphur in sediments increases with depth from 0.08 to 0.14%. The amount of total sulphur also increases with depth. This reflects an increase in the microbial decomposition of organic matter during diagenesis with depth. This confirms an increase in the intensity of bacterial sulphate reduction processes with depth. As a result of sulphate reduction, SO_4_^2−^ is reduced to H_2_S and large amounts of diagenetic pyrite (FeS_2_) are formed. The bulk of the S is fixed in the composition of pyrite. The main source of energy for these processes is the decomposition of the labile components of OM by heterotrophic microorganisms.

The chemical and mineral composition of the bottom sediments of Lake Issyk-Kul can be formed as a result of three main processes: 1. ingress of terrigenous material into the lake and dissolution of colloidal organic matter from the watershed; 2. accumulation of autochthonous OM; and 3. formation of mineral and organomineral compounds. The geochemical characteristics of the modern bottom sediments of Lake Issyk-Kul are presented in Table 2. The bottom sediments have specificity and regularity in the accumulation of Ca, Mg, Sr, Mn and Ba. This is determined by the internal water conditions of the lake (high mineralization of water, alkaline pH values, etc.) which leads to the accumulation of Ca, Mg and Sr. A geochemical feature of living matter (plankton, macrophytes, algae, etc.) is the accumulation of a number of chemical elements (Ba, Mn and Si), which, when killed and subsequently destroyed, become sediment.

### 3.3. Authigenic Minerals

The main authigenic minerals in the bottom sediments of Lake Issyk-Kul according to the data of scanning electron microscopy (SEM) are shown in the Figure 4, Figure 5 and Figure 6.

According to the SEM data, a number of authigenic minerals are found in modern bottom sediments of Lake Issyk-Kul: pyrite, troilite (or pyrrhotite), calcite, siderite, Ca-rhodochrosite, barite and amorphous silica (Figure 4, Figure 5 and Figure 6). The following terrigenous minerals are also found: quartz, chlorite, plagioclase, feldspar, monocyte, biotite, muscovite, zircon, calcium and iron phosphates. In the samples of sampling point IK-4, Cr-Ni intermetallic compounds are found, which, apparently, are of a technogenic nature.

A distinctive feature of the sediments of Lake Issyk-Kul is the large amount of carbonates, amorphous silica and framboidal pyrite. The presence of this series of minerals in the bottom sediments is associated with the chemical composition of lake waters and their increased salinity. This, with the participation of a number of microorganisms (sulphate-reducing, heterotrophic, etc.), may lead to the formation of authigenic calcite, barite and pyrite. The activity of heterotrophic organisms leads to a decrease in pH in bottom sediments, resulting in the precipitation of amorphous Si.

### 3.4. Microbial Diversity Analysis in Bottom Sediments

In the analysis of 16S rRNA gene sequences from sediment samples, it is found that more than 80% of genus-level representatives belong to undescribed organisms for which the databases used did not contain any information. Information on the diversity at the phyla and family level is presented in (Appendix A and Figure 7). The dominant phyla in all samples are *Proteobacteria*, *Chloroflexi*, *Bacteroidota*, *Verrucomicrobiota*. *Actinobacteriota* and *Firmicutes* also dominated in sample IK-3. Representatives of the phylum Desulfobacterota capable of sulphate reductions are found in all samples. *Cyanobacteria* are detected in samples 1–3. *Planctomycetota* and *Nitrospirota* are two taxa found in all samples that contribute to the oxidation of reduced forms of nitrogen. Up to 10%, depending on the phylum, belong to undescribed organisms.

At the family level, undescribed taxa average 20–30%, depending on the sample (Appendix A). The dominant family is the purple sulphur bacteria *Chromatiaceae*, chemolithoautotrophs or chemoorganoheterotrophs capable of anoxygenic photosynthesis, some of them also capable of producing elemental sulphur by sulphide oxidation [34]. Chemoheterotrophic bacteria of the family *Anaerolineaceae*, found in marine sediments [35] and capable of fermenting sugars and protein compounds of detritus, are found in all samples [36].

In samples IK-1, IK-2 and IK-4, about 5% of all OTUs belong to members of the family *Desulfosarcinaceae* (Appendix A), sulphate-reducing bacteria, typical inhabitants of marine sediments [37]. Sample IK-2 contains sulphate-reducing bacteria of the family *Desulfocapsaceae*. It should be noted that few species of the family *Desulfocapsaceae* are capable of growing chemolithoautotrophically by disproportionation of partially oxidised sulphur compounds [38].

A number of genera in the family *Spirochaetaceae* capable of dissimilative sulphate reduction [39] and *Chloroflexi* [40] are found in all samples. In addition, most microorganisms are capable of sulphate reduction in assimilative sulphate reduction processes [41].

Sample IK-3 contains OTU representatives of the chemo-organo-heterotrophic bacteria families *Arenicellaceae*, *Bacillaceae*, *Steroidobacteraceae* and *Planococacceae*, which are scarce in other samples.

Sample IK-4 had the greatest variation, with a predominance of organotrophic bacteria of enzymatic metabolic type, among which are *Hungateiclostridiaceae* and *Anaerolinaceae* [42], which have been found in hyper-arid deserts [43]. The presence of *Omnitrophicaeota* representatives is also noted in this specimen, among which acetoclastic methanogens are known [44].

Phototrophic bacteria *Chloroflexi*, *Acidobacteria*, purple sulphur bacteria *Chromatiaceae* and *Cyanobacteria*, are noted in all sediment samples. Most of them are capable of oxygenic and anoxygenic photosynthesis, thus, they may contribute to organic carbon accumulation.

The diversity indices of IC-5 samples are presented in Appendix A. According to the diversity indices, the microbial communities studied are generally similar. The highest number of OTU is found in sample IK-5 and the lowest in samples IK-3 and IK-4. According to the Simpson diversity index, all samples have a close high diversity. The lowest community equivalence is observed in samples IK-4 and IK-5.

## 4. Discussion

The use of thermodynamic modeling techniques makes it possible to evaluate the conditions for the formation and dissolution of mineral phases based on the saturation index under abiotic conditions. Phase formation is possible when the saturation index is greater than 0.5.

In this case, the use of this approach can help to establish the origin of the phase: sedimentary (chemical deposition) or formation directly in bottom sediments. The results of thermodynamic modeling are shown in Table 3.

In the aqueous phase, the formation of barite (IK-1, IK-2, IK-4 and IK-5), carbonates (cerrusite, magnesite, witherite and strontioite) is possible. At the same time, conditions for the formation of calcite and aragonite are not predicted. In a number of samples, the formation of strengite (IK-1, IK-3 and IK-5) and oxidized forms of iron (goethite, magnetite and hematite) is possible, provided there is a sufficient concentration of iron in the system. Conditions in the bottom sediments are suitable for the dissolution of barite (IK-2, IK-4) carbonate phases (rhodochrosite in all samples) and strengite. Depending on the sample, complete and partial dissolution of oxidized ferrous phases (hematite, magnetite and goethite), and for samples IK-1, IK-2 and IK-4, the formation of sulphide ferruginous minerals is predicted.

The processes of microbial decomposition of organic matter in the bottom sediments of Lake Issyk-Kul lead to a decrease in the redox potential to negative values (Figure 2). This is due to the bacterial consumption of oxygen and formation of H_2_S. The main group of microorganisms involved in organic matter decomposition may be *Chloroflexi* and *Hungateiclostridiaceae*, oxidising TOC, anaerobic chemoheterotrophs and representatives of the family *Anaerolineaceae*. The decrease in pH values in the upper sediment intervals is a consequence of the breakdown of OM under aerobic conditions. This contributes to the release of CO_2_ (due to decarboxylation of amino acids), organic acids and primary amines, resulting in acidification of the environment. The microbial decomposition of OM leads to the enrichment of pore waters with Si, PO_4_^3−^, HCO_3_^−^ and DOC (Figure 3). As a result, the most labile components of the OM are transferred to the liquid phase of the sediment and reductive environmental conditions are formed.

The microbiological processes of the carbon cycle have a major impact on the calcium cycle. In the saline Lake Issyk-Kul the calcium content is significantly higher than in freshwater ecosystems. The microbial production of CO_2_ as a result of organic matter consumption leads to the deposition of carbonate minerals in the lake. This process is particularly intensive under anaerobic conditions. The deposition of carbonates is affected by the reduction of sulphates. It is accompanied by an alkalisation of the environment due to the removal of hydrogen ion during the formation of H_2_S [2]. This leads to the formation of authigenic calcites and iron sulphides in the bottom sediments of Lake Issyk-Kul in all of the sampling points (Figure 4 and Figure 5). The authigenic (not clastogenic) origin of detected calcites is indicated by their morphology and shape. They are small (a few microns), amorphous and flocculent accumulations. Data of geochemical modeling without consideration of microorganisms contribution showed the impossibility of authigenic carbonate-Ca precipitation under given environmental conditions (Table 3). This may indicate that different groups of microorganisms, such as *Planctomycetota*, *Nitrospirota*, *Hungateiclostridiaceae* and *Anaerolinaceae*, *Desulfobacterota*, *Desulfosarcinaceae* and *Desulfocapsaceae* are involved in carbonate formation. Furthermore, authigenic CaCO_3_ deposition becomes possible when the pH rises as a result of photosynthesis.

The balance of inorganic forms of carbon (TIC) in organisms is established by the enzyme carbonic anhydrase (CA). This enzyme catalyses a number of important processes, from the photosynthetic assimilation of TIC by autotrophs to the release of CO_2_ during cellular respiration [45]. In nature, the TIC cycle is closely related to the calcium cycle. High pH values result in the removal of Ca^2+^ from the solution. Therefore, the main form of microbial action on calcium compounds is a local change in pH [46,47]. This process leads to a decrease in the Ca^2+^ content of the pore water compared to the lake water (Figure 2). Active microbial activity (OM decomposition) leads to a marked increase in DIC and HCO_3_^−^ in pore water (Figure 2).

Most of the observations of authigenic calcite in the sediments of Lake Issyk-Kul are found on the surface of plant residues (Figure 4b,c) or in areas of organic matter accumulation (Figure 4b,c). The cyanobacteria found in the sediment (Appendix A) appear to be actively involved in carbonate deposition. The ability of chemoautotrophic bacteria and cyanobacteria to precipitate calcium is well described in the literature [47,48,49]. The photosynthetic uptake of bicarbonate is associated with alkalinization of the pericellular layer, leading to the formation of CaCO_3_ granules on the glycocalyx of cyanobacteria in the presence of free Ca^2+^ ions in water [46]. The role of CA here is reduced to stabilization of pH in the pericellular space and the maintenance of substrate concentration (HCO_3_^−^), necessary both for photosynthesis and for CaCO_3_ precipitation.

The processes of the microbial decomposition of OM in lake bottom sediments proceed in parallel with bacterial reduction of sulphate. The increase of reduced forms of sulphur in the sediment of point IK-1 (Table 1) reflects the increase of the degree of decomposition of OM in the depth of the bottom sediments. This is the process of microbial mineralization of organic matter in diagenesis. In this process, the content of SO_4_^2−^ in pore water decreases compared to lake water (Figure 2). This indirectly indicates intensive bacterial sulphate-reduction processes. As a result of this process, SO_4_^2−^ is reduced to H_2_S and large amounts of authigenic pyrite are formed (Figure 5). This pyrite contains the bulk of S. The possible formation of authigenic iron sulphides in the lake is supported by the data from physicochemical modelling (Table 3). The main source of energy for these processes could be the decomposition of labile components of OM by heterotrophic microorganisms—*Proteobacteria*, *Verrucomicrobiota*, *Hungateiclostridiaceae* and *Anaerolinaceae* (Figure 7 and Appendix A).

The direction of the terminal decomposition of organic matter (sulphatreduction and methanogenesis) is controlled by the sulphate content. Results of laboratory studies have shown that at sulphate content in the medium of 11.5–21.1 mg/L sulphatreduction does not suppress methanogenesis, which develops at a high rate [50,51]. In lakes with a sulphate content of about 84–718 mg/L and above, sulphatreduction is the only final process of OM degradation [52]. In the saline Lake Issyk-Kul, the leading process of terminal decomposition of organic matter is sulphatreduction, which predominates over methanogenesis. Although acetoclastic methanogens were found in bottom sediments, representatives of the phylum of bacteria of the reducing branch of the sulphur cycle dominate in all samples: *Desulfobacterota*, *Desulfosarcinaceae* and *Desulfocapsaceae* (Figure 7 and Appendix A).

Changes in the sulphate concentration allow indirect judging of the intensity of the microbial processes of the sulphur cycle, both oxidative and reductive [53]. The bacterial reduction of sulphate ions in the pore water in the sampling locations IK-1, IK-2 and IK-4 resulted in a decrease of SO_4_^2−^ concentration and an increase of reduced sulphur forms (Figure 2; Table 1). The slight increase of SO_4_^2−^ in pore water at sampling sites IK-3 and IK-5 may be related to the oxidation process of H_2_S and Fe monosulphides by purple sulphur bacterium *Chromatiaceae* (Appendix A). It is possible that their development may be activated by access to light and may direct the biogeochemical cycle of sulphur towards oxidation. Active processes of bacterial sulphate reduction lead to the formation of large quantities of pyrite framboids in bottom sediments (Figure 5). The biogenic origin of framboid pyrite can be confidently asserted [54]. The formation of framboid structures is associated with bacterial recycling and subsequent mineralization of organic matter [55,56].

The pattern of sulphate ion distribution in bottom sediment sections affects the geochemistry of a number of elements, such as Ba. In the bottom sediments of Lake Issyk-Kul, according to SEM data, at sampling points IK-1, IK-2 and IK-3, the presence of authigenic barite is detected (Figure 6a). Barite is observed as small particles of 1–3 microns in the organic matrix of sediment. Results of physicochemical modelling show the possibility of authigenic barite formation in a number of sampling points (Table 3).

The source of the Ba entering the sludge can vary. It can be potassium containing minerals and carbonates, various silicates, clastogenic material, terrigenous barite and organic matter [57]. Most organisms (plankton, shells, radiolarians, etc.) contain barium [58]. They are a potential source of biogenic barium input to bottom sediments during their demise. Biogenic barium is “more labile” than terrigenous barium [59]. It can be easily extracted from OM in the process of its decomposition by microorganisms (e.g., heterotrophs) and get into pore water, participating in the formation of authigenic barium. Barium released during the decomposition of organic matter can create microreactive environments supersaturated with respect to barite, causing the deposition of authigenic barite in these areas [60,61].

The active development of plankton in Lake Issyk-Kul [62] may be the source of large amounts of labile Ba in bottom sediments. It is then involved in the formation of authigenic barite with the participation of microorganisms—heterotrophic bacteria (*Proteobacteria*, *Chloroflexi*, *Bacteroidota*, *Verrucomicrobiota*) that participate in the decomposition of organic matter of plankton. The mechanism of the deposition of authigenic barite in sedimentary rocks is similar to that of Fe-Mn nodules. However, barite is deposited not on the boundary of redox processes, where Fe (II) and Mn (II) are oxidized and precipitate, but on the boundary of the sharp decrease of sulphate-ion content in pore water. Usually [58,59,63] the zone of diagenetic precipitation of barite is located just below the redox boundary where active processes of bacterial sulphate reductions begin. In our case, the redox boundary is located directly at the water-sediment site. Barite is found in the uppermost intervals of the sediment (5–10 cm).

We propose the following mechanism for the formation of authigenic barite. In pore waters of deep sedimentary intervals, the concentration of SO_4_^2−^ ion decreases sharply, being consumed in the process of sulphate reductions. At the same time, Ba^2+^ ions are accumulated due to the destruction of Ba-containing OM or the dissolution of BaSO_4_ (when SO_4_^2−^ concentration decreases below the barite saturation limit). Dissolved barium diffuses into the upper intervals of the sediment and interacts with pore water containing higher concentrations of SO_4_^2−^. This process results in the formation of authigenic barite. This process leads to a decrease in the concentration of Ba^2+^ in the pore water, which is consumed in the formation of BaSO_4_. This phenomenon is observed only at those points (IK-1 and IK-2) where barite is detected by SEM (Table 3, Figure 3 and Figure 6a). Thus, processes of authigenic barite formation are closely connected with activity of sulphate-reducing bacteria.

In the sediments of Lake Issyk-Kul, silica can accumulate in three main ways. 1. Entering the sediment as a component of the terrigenous form of quartz, aluminosilicates, etc. 2. Formation of amorphous (authigenic) silica when the concentration of silica in the solution increases or when the pH is reduced to acidic values. 3. Entry of silica into bottom sediments in the form of diatom armatures. The activity of heterotrophic microorganisms in bottom sediments can lead to the deposition of amorphous silica due to anaerobic decomposition of OM, which reduces the alkalinity of lake water, thereby reducing silicon solubility. The alkaline nature of the pore water leads to a maximum degree of constriction and concentration of silica, which becomes chemically mobile. Thus, the silica valves of diatoms dissolve and dissolved Si can diffuse through the porous and water-saturated sediment to accumulation centres, where it forms an amorphous solid phase (Figure 6). Such centres can be microsites of sediment with low pH, such as organism remains and other accumulations of OM, whose decomposition produces organic acids and H_2_CO_3_, which can lead to the deposition of amorphous silica (Figure 6).

## 5. Conclusions

The basis of microbial diversity of modern bottom sediments in the littoral zone of Lake Issyk-Kul consists of representatives of *Proteobacteria*, *Chloroflexi*, *Bacteroidota* and *Verrucomicrobiota* taxa. For the Pokrov Bay with desalinated water, *Actinobacteriota* predominates in the bottom sediment sample. The dominant groups of microorganisms involved in OM decomposition are *Chloroflexi* and *Hungateiclostridiaceae*, oxidizing TOC, anaerobic chemoheterotrophs and representatives of the family *Anaerolineaceae*. Taxa providing oxidation of reduced forms of nitrogen, *Planctomycetota* and *Nitrospirota*, are noted in all samples. *Desulfobacterota*, *Desulfosarcinaceae* and *Desulfocapsaceae* are the dominant microbial groups of the reducing branch of the sulphur cycle. Representatives of the oxidative branch of sulphur are the purple sulphur bacteria *Chromatiaceae*.

The high abundance of all indicator groups of microorganisms indicates the presence of labile organic components in bottom sediments, despite the low TOC content (0.2–1.9%). The OM degradation processes in the bottom sediments are quite active. This leads to an increase of DOC, HCO^3−^ and PO_4_^3−^ concentration in the pore water, a decrease of pH values and the formation of reducing environment conditions. This leads to the accumulation of reduced forms of sulphur and a number of chemical elements (Ca, Fe, Ba, Sr, etc.) in the bottom sediments. The microbial community in the biotope of the littoral zone of Lake Issyk-Kul changes the physico-chemical conditions of the bottom sediments (decrease in pH and Eh) and takes an active part in the formation of various authigenic minerals. The energy for these processes is the microbial decomposition of organic matter.

Based on biogeochemical studies, we assume the participation of different physiological groups of microorganisms in the formation of authigenic minerals (microorganisms necessary for these processes inhabit bottom sediments):i.The activities of heterotrophic and sulphate-reducing microorganisms may produce authigenic carbonate minerals (calcite);ii.The activity of sulphate-reducing bacteria may provide the basis for the growth of reduced sulphur forms and the formation of authigenic sulphide minerals (rambogenic pyrite);iii.The activity of sulphate-reducing bacteria can lead to the deposition of barite at the redox boundary;iv.The activity of heterotrophic microorganisms can lead to a decrease in pH (due to the decomposition of organic matter), which creates conditions for the deposition of amorphous silica.

The described mechanisms of microbial mineral formation require further experimental confirmation and clarification, which will be the subject of our further research.

## Figures and Tables

**Figure 1 biology-12-00642-f001:**
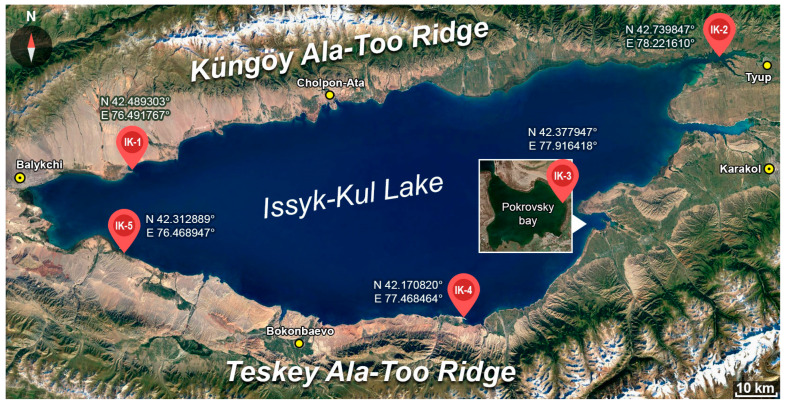
Sampling points and coordinates.

**Figure 2 biology-12-00642-f002:**
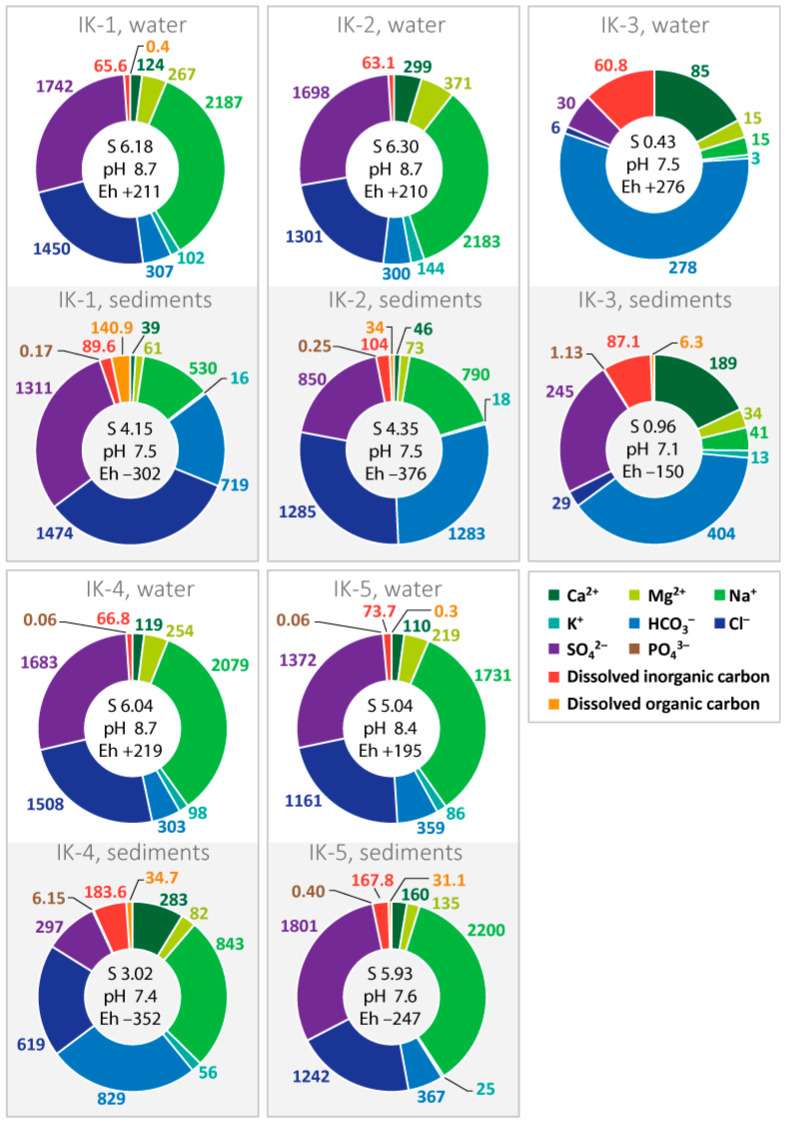
Main ionic composition (mg/L), salinity (S) (g/L), carbon forms (g/L), pH, Eh (mV) of lake water and pore water of bottom sediments of Lake Issyk-Kul.

**Figure 3 biology-12-00642-f003:**
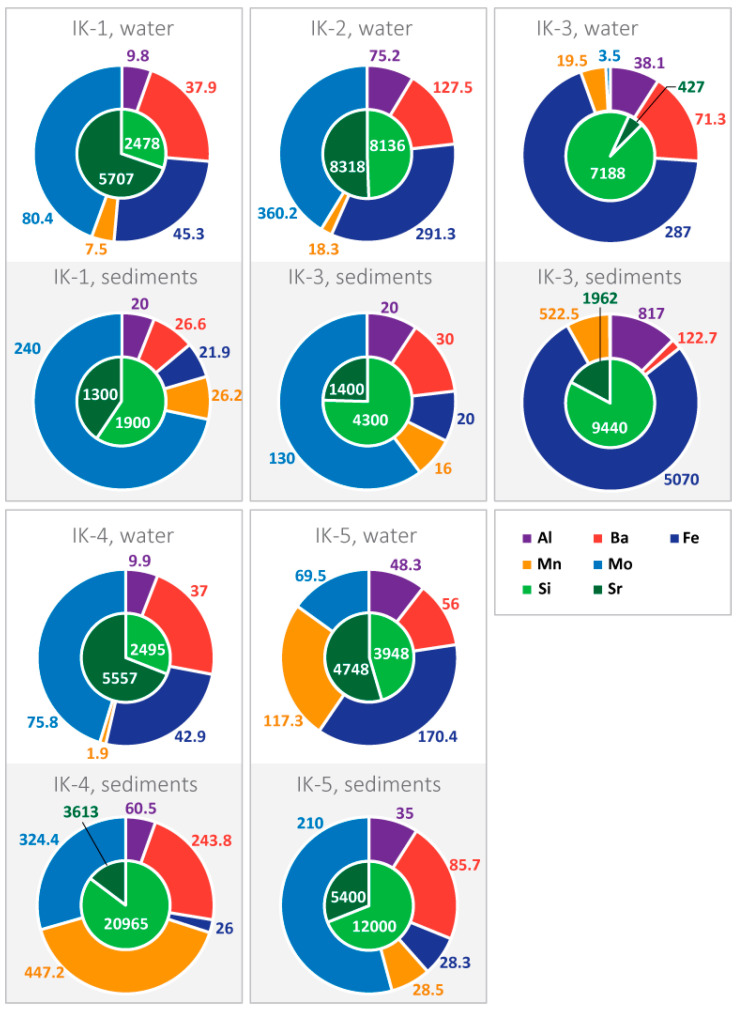
Elemental composition (µg/L) of lake water and pore water of bottom sediments of Lake Issyk-Kul.

**Figure 4 biology-12-00642-f004:**
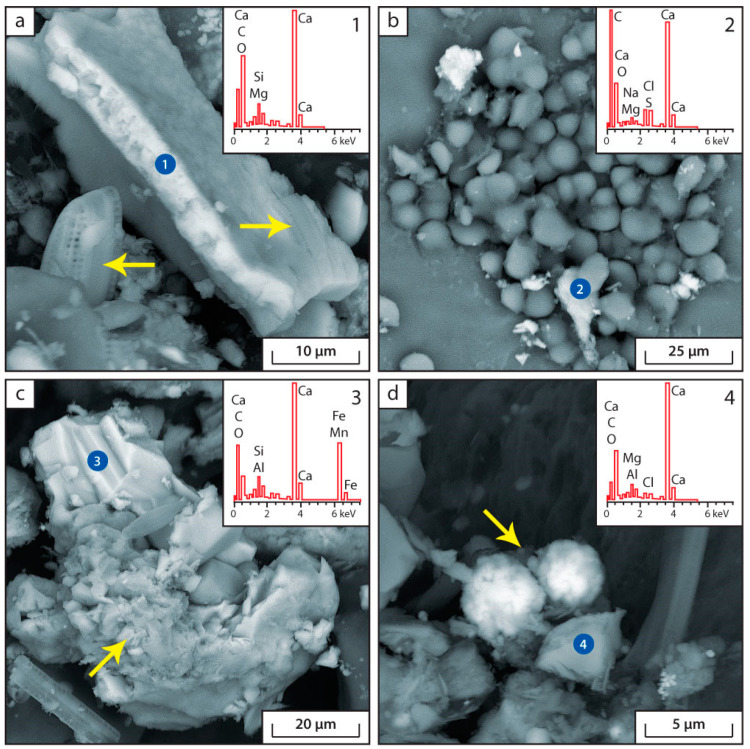
Microphotographs and energy dispersion spectra of authigenic carbonates in bottom sediments. (**a**) Point IK-1, pseudomorphs of calcite after plant remains (arrows indicate diatoms); (**b**) Point IK-2, flaky formations of calcite on the surface of Chara algae (oolites—organic formations rich in Ca); (**c**) Point IK-2 is a solid solution of siderite and Ca-rhodochrosite in the organic matrix of the sediment (the arrow shows amorphous silica); (**d**) Point IK-4, newly formed calcite in the form of flakes and pseudomorphs after plant remains (the arrow shows pyrite framboids).

**Figure 5 biology-12-00642-f005:**
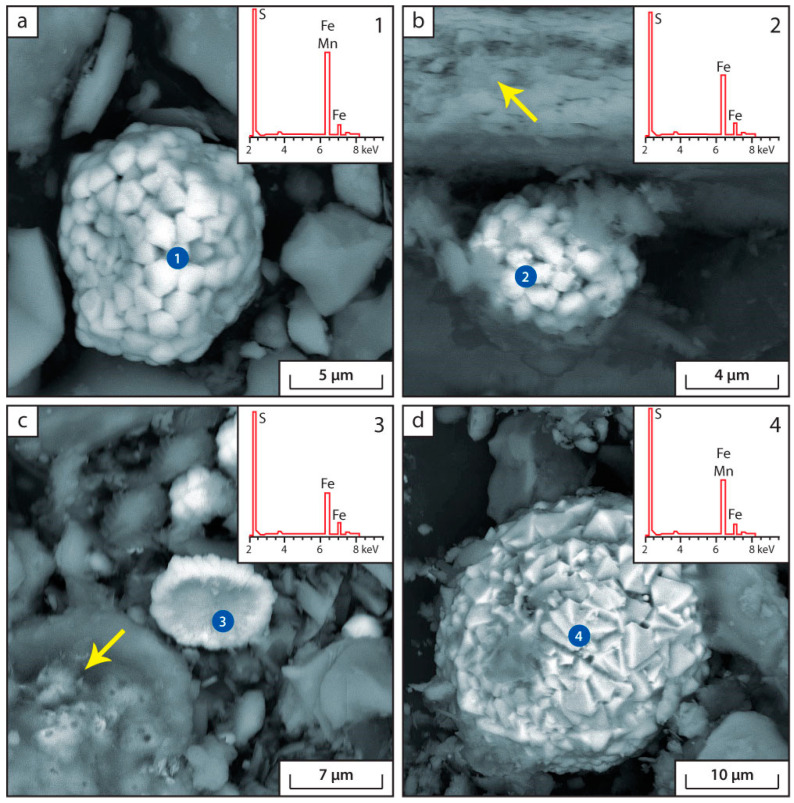
Microphotographs and energy dispersion spectra of authigenic pyrite in bottom sediments. (**a**) Point IK-1, framboidal pyrite with inclusion of Mn (up to 1%); (**b**) Point IK-3, framboidal pyrite (arrow shows plant remains enriched in Ca); (**c**) Point IK-4, framboids and pyrite flakes (arrow shows oolites of native Fe in organic matrix); (**d**) Point IK-5 is a framboidal pyrite.

**Figure 6 biology-12-00642-f006:**
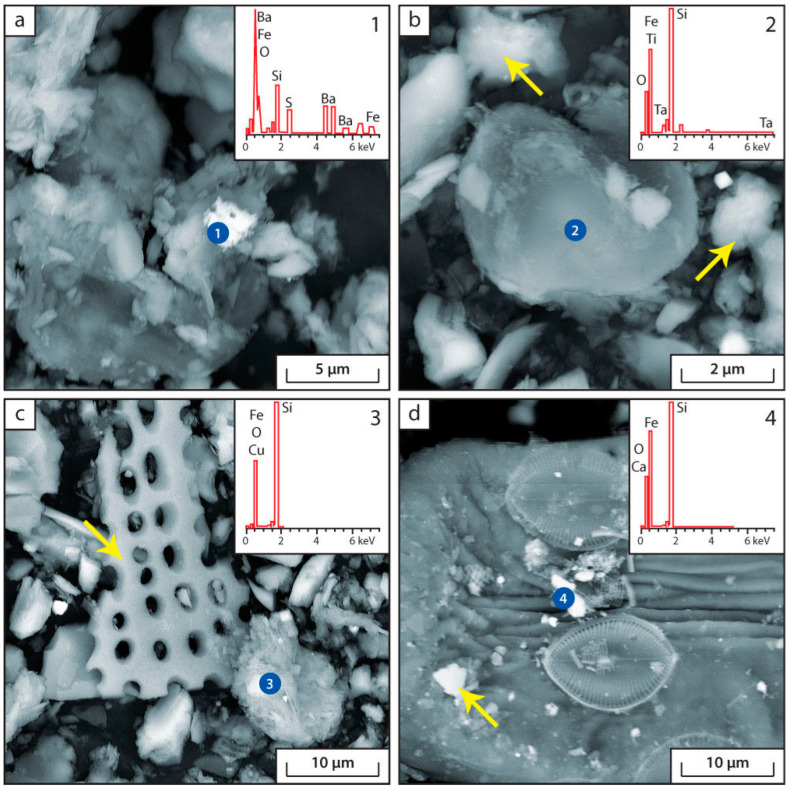
Microphotographs and energy dispersion spectra of authigenic minerals in bottom sediments. (**a**) Point IK-3, microparticles of authigenic barite among flocculent particles of amorphous silica; (**b**) Point IK-1 is a microparticle of amorphous silica (arrows show flaky particles of authigenic calcite); (**c**) Point IK-2 is a microparticle of amorphous silica (arrow shows a diatom valve); (**d**) Point IK-5 is a microparticle of newly formed silica among diatom valves (the arrow shows a flocculent particle of authigenic calcite).

**Figure 7 biology-12-00642-f007:**
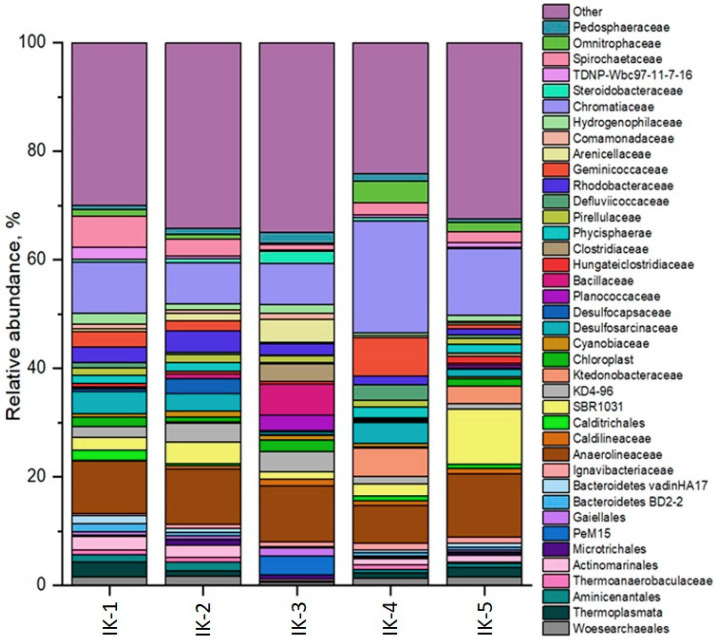
Phylogenetic diversity of bottom sediment microbial communities based on the analysis of 16S rRNA gene sequences at the familia level.

**Table 1 biology-12-00642-t001:** Content of total carbon (TC), total organic carbon (TOC), H, N, and forms of sulphur (%) in bottom sediments.

Sample/Depth	TC	TOC	H	N	S Total	S (VI)	S (II)
IK-1, 0–5 cm	2.56	0.9	<0.3	<0.3	0.11	0.03	0.08
IK-1, 25–30 cm	1.66	0.2	<0.3	<0.3	0.15	0.01	0.14
IK-2	2.66	0.5	0.40	<0.3	0.19	0.02	0,17
IK-3	4.25	1.9	0.52	0.45	0.18	0.02	0.16
IK-4	1.13	0.5	<0.3	<0.3	0.26	0.08	0.18
IK-5	4.15	1.1	0.48	<0.3	0.32	0.03	0.29

**Table 2 biology-12-00642-t002:** The chemical composition of the bottom sediment.

Sample/Depth	Al_2_O_3_	CaO	Fe_2_O_3_	K_2_O	MgO	MnO	Na_2_O	SiO_2_	Ba	Mo	Sr
	%	%	%	%	%	%	%	%	ppm	ppm	ppm
IK-1, 0–5 cm	12.7	10.4	3.31	2.82	2.21	0.063	3.05	52.2	651	11.6	813.1
IK-1, 25–30 cm	13.8	7.0	3.64	2.99	2.22	0.066	3.34	65.8	687	5.5	333.1
IK-2	15.1	9.0	4.13	3.46	2.12	0.076	2.80	57.9	658	23.1	467.9
IK-3	12.0	8.3	3.96	2.61	2.55	0.072	2.20	56.8	513	4.0	542.3
IK-4	12.8	7.5	5.22	3.04	1.93	0.097	2.88	54.6	801	47.0	511.2
IK-5	11.8	14.3	3.87	2.71	3.48	0.072	1.90	45.5	581	19.8	599.8

**Table 3 biology-12-00642-t003:** Saturation indexes for mineral phases in water (W) and bottom sediments (BS).

Mineral Phase	IK-1		IK-2		IK-3		IK-4		IK-5	
	W	BS	W	BS	W	BS	W	BS	W	BS
Anhydrite CaSO_4_	−4.71	−4.71	−5.06	−13.86	−6.53	−6.53	−5.00	−10.28	−4.66	−4.66
Barite BaSO_4_ *	1.73	1.72	1.81	−6.98	−0.13	−0.13	1.75	−3.55	1.59	1.59
Aragonite CaCO_3_	−2.34	−2.34	−3.05	−3.05	−3.93	−3.93	−2.96	−2.95	−2.48	−2.48
Calcite CaCO_3_ *	−2.19	−2.19	−2.91	−2.90	−3.78	−3.78	−2.81	−2.81	−2.33	−2.33
Cerussite PbCO_3_	0.18	0.19	−0.83	−0.82	−1.84	−1.83	−0.37	−0.34	−0.50	−0.48
Magnesite MgCO_3_	0.25	−0.02	−0.67	−0.67	−1.47	−1.46	−0.34	−0.33	−0.26	−0.26
Witherite BaCO_3_	3.42	3.41	3.14	3.15	1.79	1.79	3.10	3.10	3.09	3.09
Strontianite SrCO_3_	0.97	0.97	0.31	0.31	−0.76	−0.76	0.65	0.65	0.55	0.55
Strengite FePO_4_⋅2H_2_O	0.67	−6.54	−0.29	−7.85	2.11	−4.29	−0.32	−7.22	0.19	−5.55
Fe(OH)_3_	2.49	−3.37	2.49	−5.07	2.59	−3.81	2.71	−4.17	2.59	−3.15
Goethite FeOOH	7.63	1.77	7.63	0.07	7.73	1.33	7.84	0.96	7.73	1.99
Hematite Fe_2_O_3_	16.24	0.98	16.24	−0.01	16.43	0.23	16.66	−0.12	16.44	0.23
Fe(OH)_2_	−5.18	−1.92	−5.17	−2.30	−5.07	−3.86	−5.11	−1.83	−4.50	−2.37
Magnetite Fe_3_O_4_	14.61	6.16	14.63	2.38	14.86	3.33	15.11	4.63	15.49	6.15
Pyrite FeS_2_ *	−127.21	3.87	−126.86	4.44	−125.17	−17.82	−129.22	5.74	−117.31	−5.05
Pyrrhotite FeS	−76.11	0.20	−75.90	1.61	−74.99	−13.13	−77.22	2.07	−70.23	−5.17

* minerals also detected by SEM.

## Data Availability

Not applicable.

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
