# Peer review of "Microbial Diversity and Authigenic Mineral Formation of Modern Bottom Sediments in the Littoral Zone of Lake Issyk-Kul, Kyrgyz Republic (Central Asia)"

_biology, 2023, doi:10.3390/biology12050642_

Round 1

Reviewer 1 Report (Previous Reviewer 2)

Move table 3 and 4 to Supporting Information (SI). Possible move either figure 5 or 6 to SI as well. The manuscript is overwhelmed with too many figures and tables and that can take away the actual message authors trying to disseminate.

Author Response

Move table 3 and 4 to Supporting Information (SI). Possible move either figure 5 or 6 to SI as well. The manuscript is overwhelmed with too many figures and tables and that can take away the actual message authors trying to disseminate.

Figures 2 and 7, table 3 were moved to Supplementary section

Reviewer 2 Report (New Reviewer)

The work is within the scope of the journal, although it has strong geomicrobiological/geochemical aspect

The approaches and methods are adequate for achieving the goals. A combination of classic geochemical, mineralogical and genetic tools is a big and I would say, rare, advantage.

The text is poorly written, especially the Abstract. Serious English proofread is definitely requested.

Introduction: no working hypothesis (based on what is known for this lake and other lakes of Eurasia); no specific objectives. This should be seriously revised.

Section 2.1. Repetitive wording (sand content etc)

The first paragraph of the Discussion contradicts to what is stated in section 2.2

P 17, end of 2nd paragraph: Note that this precipitation becomes possible when pH rises due to photosynthesis

4th paragraph, rewrite the first sentence, it is unclear

Last paragraph of the Discussion: microniches in sediments are not evidenced by direct measurements. What are these authigenic Si minerals? Amorphous Mg silicates? Please elaborate.

The concentration of dissolved Si does not imply a possibility of amorphous SiO2xnH2O precipitation, unless the authors can prove it be thermodynamic calculations. Could it be amorphous Mg silicates?

Fig 1: What are the units of ions, mg/L?

Author Response

The text is poorly written, especially the Abstract. Serious English proofread is definitely requested.

Abstract was seriously rewritten

Introduction: no working hypothesis (based on what is known for this lake and other lakes of Eurasia); no specific objectives. This should be seriously revised.

Introduction was revised

Section 2.1. Repetitive wording (sand content etc)

Done

The first paragraph of the Discussion contradicts to what is stated in section 2.2

Fixed

P 17, end of 2nd paragraph: Note that this precipitation becomes possible when pH rises due to photosynthesis

This sentence was added

4th paragraph, rewrite the first sentence, it is unclear

corrected

Last paragraph of the Discussion: microniches in sediments are not evidenced by direct measurements. What are these authigenic Si minerals? Amorphous Mg silicates? Please elaborate.

corrected

The concentration of dissolved Si does not imply a possibility of amorphous SiO2xnH2O precipitation, unless the authors can prove it be thermodynamic calculations. Could it be amorphous Mg silicates?

The new version is:

In bottom sediments Issyk-Kul Lake silicon can be accumulated in three main ways. 1. Receipt in the sediment as a component of terrigenous form of quartz, aluminosilicates, etc. 2. Formation of amorphous (authigenic) silica with an increase in the concentration of silicon in the solution or with a decrease in pH to acidic values. 3. Silicon entry into bottom sediments in the form of diatom valves. The activity of heterotrophic microorganisms in the sediments can lead to the precipitation of amorphous silica due to the anaerobic decomposition of OM, which reduces the alkaline values of lake water, thereby reducing the solubility of silicon. The alkaline nature of the pore waters leads to the maximum degree of constriction and concentration of silica, which becomes chemically mobile. Thus, the flint valves of diatoms dissolve and dissolved Si can diffuse through the porous and water-saturated sediment to the centers of crystallization, where it forms an amorphous solid phase (Figure 7b–d). Such centers are sediment microsites with low pH, for example, at the remains of organisms and other accumulations of OM, the decomposition of which generates organic acids and H2CO3, which can lead to the precipitation of amorphous silica (Figure 7d).

Fig 1: What are the units of ions, mg/L?

Fixed, for fig 3

Reviewer 3 Report (New Reviewer)

The manuscrit presents the biogeochemical characterization of a particular environment, the coastal sediments of a high altitude salt lake. Results show the mineralization of the organic matter and discuss the formation of minerals in relation with the putative microbial activities. The authors should be less affirmative in their discussion and conclusion section. Even if their hypothesis are highly probable, the characterization of the microbial compartment is based on DNA, so the real activities are not visible. The hypothesis should be given then the way to confirm them should be presented in the conclusion (activity tests, RNA-based analyses...). The english style is correct at the beginning of the manuscript, however it deteriorates progressively at the end of the work. It should be checked carefully.

Specific comments :

Abstract : this sentence has no verb : “Active destruc-tion of organic matter starting at the water—bottom sediment boundary.”

Page 3: verify “2-3 g/L of phosphate, really g/L and not mg?

Page 4: analytical and not analytical

Page 5 “The amount of S (II) was estimated as the difference between Stotal and S (VI).”, but you don’t explain how you analyze SVI

The legend of Figure 3 is in yellow

Table 2, you must precise more clearly the unit for each compound (% or ppm)

Page 15, « occurring in marine sediments and capable of fermentation”

“dissimilative” in place of “dissimilation”

“they can contribute to the accumulation…”

Page 17 “organic matter decomposition”

“chemoheterotrophs”

The data of “geochemical modeling”

Most “observations” and not “finds”

“At a sulfate content in the medium of 11.5–21.1 mg/L” this sentence is not suitable; you should give a larger interval, or precise “the results of a laboratory study showed that…”

Page 19: at the water-sediment interfacte, and not section

Silicon enters and not “entry”

“centers of crystallization, where it forms an amorphous solid phase”, which crystallization if it is amorphous?

Actinobacteria and not actonobacteria

Chemoheterotroph and not chemoheteratroph

Author Response

Responses to the reviewer #3

The authors should be less affirmative in their discussion and conclusion section. Even if their hypothesis are highly probable, the characterization of the microbial compartment is based on DNA, so the real activities are not visible. The hypothesis should be given then the way to confirm them should be presented in the conclusion (activity tests, RNA-based analyses...).

Taking into account your fair remark, we indicated in the text that the conclusions are made on the basis of indirect evidence and require further experimental confirmation. However, our conclusions are confirmed by geochemical data (changes in the composition of sediments, lake waters, pore waters, biogenic elements), are confirmed by scanning electron microscope, geochemical modelling, and sequencing of 16S rRNA genes. The mechanisms of microbial mineral formation described by us are confirmed by literary sources on other lakes.

The English style is correct at the beginning of the manuscript; however it deteriorates progressively at the end of the work. It should be checked carefully.

The English was checked carefully.

In the Conclusion section, we have added:

“The described mechanisms of microbial mineral formation require further experimental confirmation and refinement, which will be the subject of our further research.”

Specific comments:

Abstract : this sentence has no verb : "Active destruction of organic matter starting at the water—bottom sediment boundary."

Fixed — Active destruction of organic matter is starting at the water—bottom sediment boundary.

Page 3: verify "2-3 g/L of phosphate, really g/L and not mg?

Fixed — (~2–3.8 µg/L …

Page 4: analytical and not analytical

Fixed — Analytical Methods.

Page 5 "The amount of S (II) was estimated as the difference between Stotal and S (VI).", but you don't explain how you analyze SVI

Sulfate sulfur is sulfur S (VI). We have added this explanation to the text.

“S (II) was removed from specimens via digestion in diluted HCl and subsequent filtering of the residue, whereby only sulfate sulfur (S (VI)) remained.”

The legend of Figure 3 is in yellow

Fixed

Table 2, you must precise more clearly the unit for each compound (% or ppm)

Fixed

Page 15, "occurring in marine sediments and capable of fermentation"

"dissimilative" in place of "dissimilation"

"they can contribute to the accumulation..."

Fixed

Page 17 "organic matter decomposition"

"chemoheterotrophs"

The data of "geochemical modeling"

Most "observations" and not "finds"

Fixed

"At a sulfate content in the medium of 11.5–21.1 mg/L" this sentence is not suitable; you should give a larger interval, or precise "the results of a laboratory study showed that..."

Fixed — The results of a laboratory study showed that at a sulfate content in the medium of 11.5–21.1 mg/L, sulfate reduction does not suppress methanogenesis, which develops at a high rate .

Page 19: at the water-sediment interfacte, and not section

Silicon enters and not "entry"

Fixed

"centers of crystallization, where it forms an amorphous solid phase", which crystallization if it is amorphous?

Fixed — Thus, the flint valves of diatoms dissolve and dissolved Si can diffuse through the porous and water-saturated sediment to the centers of accumulation, where it forms an amorphous solid phase.

Actinobacteria and not actonobacteria

Chemoheterotroph and not chemoheteratroph

Fixed

Reviewer 4 Report (New Reviewer)

The authors have put together an interesting manuscript describing the chemistry and microbial diversity of littoral sediments in five locations of a saline mountain lake. There are a few places where corrections need to be made and some where more information needs to be presented for readers. 

Comments:

Title: The title 'Microbial role in authigenic mineral formation of modern bottom sediments, etc.' is not accurate. There were no microbial activity measurements made, so determining the microbial role is only an assumed one. The authors did show the diversity of the microorganisms using the 16S rRNA gene. However, this methodology will not distinguish between DNA and 16S rRNA genes from live or dead cells. Perhaps a better title would be 'Microbial diversity and authigenic mineral formation of modern bottom sediments in the littoral zone of Issyk-Kul Lake, Kyrgyz Republic (Central Asia).'

Abstract: Line 9; change sentence (Has been established...) to 'The participation of microorganisms in processes in the formation of a number of authigenic minerals (calcite, framboidal pyrite, barite, and amorphous Si) has been established.'

Introduction: (page 3) Line 12; eliminate the word 'was' and change 'role of vaterite' to 'role in vaterite'

Line 17; change to 'Pseudomonas and Micrococcus'. Also italicize the genera names.

Paragraph starting Line 30; instead of 'establish the role, etc.', change to 'relate the diversity of various physiological groups of microorganisms in the biogeochemical processess of the littoral zone (decomposition of OM, distribution of chemical elements, transformation of pore water, etc.). The rest of the sentence should be eliminated because showing direct participation in authigenic mineral formation would require some sort of activity measurement. 

Sampling: The authors state that 'In each location were taken water samples and samples from of bottom sediments for geochemical and microbiological analyses.' However, there is no indication of how water samples were collected (i.e., what kind of sampler was used) and how the sampler was treated in between locations to avoid or minimize cross contamination. Similarly, what kind of sampler was used to collect sediment samples? Was it a grab or a coring device? Were the authors able to obtain sediment samples with an intact water/sediment interface? If a coring device was used, were subsamples taken from the core? More details are necessary in this section to provide readers with an accurate picture of how collecting samples was accomplished.

Analytical (change from Analitical) Methods.

Page 6, Lines 1-2; Readers should not have to search out another article to figure out how methods were performed. A brief description of how the methods and analysis of the samples for SEM should be presented.

Page 15, Line 10; change to ' It should be noted' (not nouted).

Lines 20-21; change to 'among which were Hungateteiclostridiaceae and Anaerolinaceae [42] that have been found in hyper-arid deserts [43].'

Line 25; change to 'thus, they can contribute'

Page 20: The last paragraph of the Conclusions needs to be completely rewritten. The authors state they have 'established the participation of different physiological groups of microorganisms in the formation of the following minerals:' when they have not done so. In order to establish the direct participation of the microbes in these processes, some sort of microbial activity measurements in the formation would be required. Yes, they have established that the right microorganisms for these processes are indeed present, but for this statement to be true, they need to back up the presence of these microorganisms with tests to show they are actively involved in the formation of the authigenic minerals.

Author Response

Responses to the reviewer #4

Dear Reviewer, Thank you for your valuable comments on our manuscript. All of them have been taken into account in the text.

Title: The title 'Microbial role in authigenic mineral formation of modern bottom sediments, etc.' is not accurate. There were no microbial activity measurements made, so determining the microbial role is only an assumed one. The authors did show the diversity of the microorganisms using the 16S rRNA gene. However, this methodology will not distinguish between DNA and 16S rRNA genes from live or dead cells. Perhaps a better title would be 'Microbial diversity and authigenic mineral formation of modern bottom sediments in the littoral zone of Issyk-Kul Lake, Kyrgyz Republic (Central Asia).'

Fixed

Abstract: Line 9; change sentence (Has been established...) to 'The participation of microorganisms in processes in the formation of a number of authigenic minerals (calcite, framboidal pyrite, barite, and amorphous Si) has been established.'

Fixed

Introduction: (page 3) Line 12; eliminate the word 'was' and change 'role of vaterite' to 'role in vaterite'

Fixed

Line 17; change to 'Pseudomonas and Micrococcus'.

Fixed

Also italicize the genera names.

Fixed

Paragraph starting Line 30; instead of 'establish the role, etc.', change to 'relate the diversity of various physiological groups of microorganisms in the biogeochemical processess of the littoral zone (decomposition of OM, distribution of chemical elements, transformation of pore water, etc.). The rest of the sentence should be eliminated because showing direct participation in authigenic mineral formation would require some sort of activity measurement.

Sampling: The authors state that 'In each location were taken water samples and samples from of bottom sediments for geochemical and microbiological analyses.' However, there is no indication of how water samples were collected (i.e., what kind of sampler was used) and how the sampler was treated in between locations to avoid or minimize cross contamination. Similarly, what kind of sampler was used to collect sediment samples? Was it a grab or a coring device? Were the authors able to obtain sediment samples with an intact water/sediment interface? If a coring device was used, were subsamples taken from the core? More details are necessary in this section to provide readers with an accurate picture of how collecting samples was accomplished.

Fixed

Analytical (change from Analitical) Methods.

Fixed

Page 6, Lines 1-2; Readers should not have to search out another article to figure out how methods were performed. A brief description of how the methods and analysis of the samples for SEM should be presented.

Fixed

Page 15, Line 10; change to ' It should be noted' (not nouted).

Fixed

Lines 20-21; change to 'among which were Hungateteiclostridiaceae and Anaerolinaceae [42] that have been found in hyper-arid deserts [43].'

Fixed

Line 25; change to 'thus, they can contribute'

Fixed

Page 20: The last paragraph of the Conclusions needs to be completely rewritten. The authors state they have 'established the participation of different physiological groups of microorganisms in the formation of the following minerals:' when they have not done so. In order to establish the direct participation of the microbes in these processes, some sort of microbial activity measurements in the formation would be required. Yes, they have established that the right microorganisms for these processes are indeed present, but for this statement to be true, they need to back up the presence of these microorganisms with tests to show they are actively involved in the formation of the authigenic minerals.

Fixed

Round 2

Reviewer 2 Report (New Reviewer)

The authors adequately revised the manuscript and adressed most reviewer's comments. Still the problem remains with units of ion concentration in Fig 2 - it cannot be g/L, rather mg/L. Please check.

Author Response

Dear Reviewer, thank you very much for taking the time to read our manuscript and for your important comments. Indeed, there was a typo in the text, ion concentrations are in mg/l, organic matter and salt content in g/l. We have rechecked English and corrected the text.

Reviewer 4 Report (New Reviewer)

The authors have made the requested changes. There are no further changes requested.

Author Response

Dear Reviewer, thank you very much for taking the time to read our manuscript and for your important comments.

This manuscript is a resubmission of an earlier submission. The following is a list of the peer review reports and author responses from that submission.

Round 1

Reviewer 1 Report

Dear authors,

The manuscript with the title “Microbial diversity and biogeochemical authigenic mineral formation in the bottom sediments of Issyk-Kul Lake, Kyrgyz Republic (Central Asia)” is presented as a multidisciplinary study, in which geochemical, physico-chemical and microbial analyses are used to ultimately understand the role of microbial activity in the formation of authigenic minerals in the study area.

I find the analyses performed sufficient and appropiate for this study. However, I detect an important and serious flaw in the microbial analysis and results. Methods are not well explained, and results are misunderstanding and sometimes even wrongly explained. Thus, the manuscript loses consistency and meaning, and makes difficult to reader to follow the explanation on how the microorganisms interfere in the formation of authigenic minerals.

The study has potential to contribute to a better understanding of biomineralizations processes, but the manuscript needs to be improved.

General comments:

- A graphical abstract is useful, but I am not sure if using images from the internet (framboidal pyrite image, plankton image, sulfate molecule image, etc) or other sources is appropriate or needs references.

- Please, be consistent with the name of the studied lake and always use the same name. In this case, I would suggest always using Issyk-Kul Lake.

- Pay attention to characters that must be written as subscript or superscript (e.g km2 and not km2).

- Please, be consistent with the use of en-dash, em-dash and minus along the manuscript.

- Citing or referring to other studies like “according to [number]” or “In study [number]” is not appropriate.

- Please, check the .pdf file since there may be additional comments there.

- Change "bacteria destructors" to "heterotrophic bacteria", "bacterial heterotrophs" or "bacterial decomposers". It is not common to use bacterial destruction or “the destruction of organic matter by bacteria”.

-While methods and results from the physico-chemical geochemical analyses are acceptable and correctly expressed among the manuscript, methods and results regarding microbial analyses lack of consistency, information in the text is very confusing and some data is wrongly explained.

SIMPLE SUMMARY

- Change "bacteria destructors" to "heterotrophic bacteria", "bacterial heterotrophs" or "bacterial decomposers".

- Change "the organic matter aerobic and anaerobic oxidation" to "aerobic and anaerobic oxidation of organic matter...".

ABSTRACT

-Change “in the world - Issyk-Kul (along the entire perimeter of the lake)” to “in the world, the Issyk-Kul Lake”.

- Change “organic carbon destructors” to “organic carbon decomposers”

- Change “Verrukomicrobiota” to “Verrucomicrobiota”

- In the sentence „…and bacteria of the oxidizing and reducing branches of the biogeochemical sulfur cycle (representatives of Desulfobacterota, Desulfosarcinaceae and Desulfocapsaceae)”, the taxa you mentioned include only sulfate-reducers and sulfur-oxidizers are missing.

- Consider rephrasing “The high diversity and uniformity of groups of microorganisms in communities indicates the presence in sediments of labile organic components involved in modern biogeochemical processes, with active destruction of organic matter starting at the water-sediment boundary”.

INTRODUCTION

- Delete “It is the second-largest mountain lake in the world behind Lake Titicaca in South America”.

- Change “keep” to “keeps”.

- Change “In study [5] authors…” to, for example, “In 2021, Rojas-Jimenez and colleagues [5]…”

- Delete “A” from “A depth and temperature…”

- Delete “the” from “…with the variations in Cyanobacteria”.

- In the sentence, “The dominance of Planctomycetes and Chloroflexi in the deepest layers can only be seen in a few lakes of the world”. Is that true? If so, references are required.

- Change “in bottom sediments for Issyk-Kul” to “in bottom sediments of Issyk-Kul Lake”.

- Change “in comparison with” to “in comparison to”.

- Rephrase “In study [10]…”

- Delete comma after i.e.

- In the sentence “The photosynthetic activity at the bottom of the lake, […], seem to trigger the epitaxial precipitation of the oriented fibers of vaterite”, why these findings are relevant for this study?

- Issyk-Kul should always come with the word “Lake”.

- Delete “Issyk-Kul is an important center of tourism and recreation in Kyrgyzstan. However,”. Further in the same paragraph, why is the ecosystem of Issyk-Kul Lake fragile? This paragraph also needs to be rephrased.

- Paragraph starting with “Of greatest concern is mining and its waste” also needs to be rephrased. Consider including only what is relevant for this study.

- Delete dot in “Issyk-Kul Lake, to study…”

PHYSIOGRAPHIC BACKGROUND AND METHODS

- Change “degrees Celsius” to “ºC”.

- Change “monomietie”. What is it in English?

- Change mg/I to mg/l (with an L and not with an I). It is miligrams/liter.

- Figure1: consider changing the sampling points to IK-1, IK-2, IK-3… as in Table1 and in the text.

- Change Stotal to Stotal, S(VI) to SO42-, and S (II) to S2+.

- The part where you explain environmental 16S rRNA gene analysis is not well described and text is missing. Also, references are missing regarding the preparation of libraries. Subsections are missing as well, since there is a subsection titled “geomechical modelling”.

-What is llnl?

RESULTS

- Delete “active” in “active microbial activity”.

- In subsection 3.3., why do you make the statement “all of them are organotrophic bacteria”? At that taxonomic level, you cannot conclude that all Proteobacteria or Verrucomicrobiota (for example) are organotrophic. Do you have further data to support that conclusion?

- Rephrase “Representatives of the bacterial phylum of the reducing branch of the sulfur cycle Desulfobacterota were found in all samples”.

- Change “oxidization” to “oxidation”.

- Rephrase paragraph “The second most representative family…”.

- Change “Spirohaetaceae” to “Spirochaetaceae”.

- In the paragraph “Green bacteria observed in bottom sediment samples…”, then you refer to phototrophic Acidobacteria, PURPLE sulfur bacteria, etc. Are you saying purple sulfur bacteria are green bacteria?

- What is a diversity index? How did you measure it?

DISCUSSION

Results need to be well understood to consider the peer-review of the discussion.

Author Response

Reply to REW 1

Dear authors,

The manuscript with the title “Microbial diversity and biogeochemical authigenic mineral formation in the bottom sediments of Issyk-Kul Lake, Kyrgyz Republic (Central Asia)” is presented as a multidisciplinary study, in which geochemical, physico-chemical and microbial analyses are used to ultimately understand the role of microbial activity in the formation of authigenic minerals in the study area.

I find the analyses performed sufficient and appropiate for this study. However, I detect an important and serious flaw in the microbial analysis and results. Methods are not well explained, and results are misunderstanding and sometimes even wrongly explained. Thus, the manuscript loses consistency and meaning, and makes difficult to reader to follow the explanation on how the microorganisms interfere in the formation of authigenic minerals.

The study has potential to contribute to a better understanding of biomineralizations processes, but the manuscript needs to be improved.

General comments:

- A graphical abstract is useful, but I am not sure if using images from the internet (framboidal pyrite image, plankton image, sulfate molecule image, etc) or other sources is appropriate or needs references.

Fixed (We used only our own pictures of minerals).

- Please, be consistent with the name of the studied lake and always use the same name. In this case, I would suggest always using Issyk-Kul Lake.

Fixed

- Pay attention to characters that must be written as subscript or superscript (e.g km2 and not km2).

Fixed

- Please, be consistent with the use of en-dash, em-dash and minus along the manuscript.

Fixed

- Citing or referring to other studies like “according to [number]” or “In study [number]” is not appropriate.

Fixed

- Please, check the .pdf file since there may be additional comments there.

- Change "bacteria destructors" to "heterotrophic bacteria", "bacterial heterotrophs" or "bacterial decomposers". It is not common to use bacterial destruction or “the destruction of organic matter by bacteria”.

Fixed

-While methods and results from the physico-chemical geochemical analyses are acceptable and correctly expressed among the manuscript, methods and results regarding microbial analyses lack of consistency, information in the text is very confusing and some data is wrongly explained.

SIMPLE SUMMARY

- Change "bacteria destructors" to "heterotrophic bacteria", "bacterial heterotrophs" or "bacterial decomposers".

Fixed

- Change "the organic matter aerobic and anaerobic oxidation" to "aerobic and anaerobic oxidation of organic matter...".

Fixed

ABSTRACT

-Change “in the world - Issyk-Kul (along the entire perimeter of the lake)” to “in the world, the Issyk-Kul Lake”.

Fixed

- Change “organic carbon destructors” to “organic carbon decomposers”

Fixed

- Change “Verrukomicrobiota” to “Verrucomicrobiota”

Fixed

In the sentence „…and bacteria of the oxidizing and reducing branches of the biogeochemical sulfur cycle (representatives of Desulfobacterota, Desulfosarcinaceae and Desulfocapsaceae)”, the taxa you mentioned include only sulfate-reducers and sulfur-oxidizers are missing.

Fixed

- Consider rephrasing “The high diversity and uniformity of groups of microorganisms in communities indicates the presence in sediments of labile organic components involved in modern biogeochemical processes, with active destruction of organic matter starting at the water-sediment boundary”.

Fixed

INTRODUCTION

- Delete “It is the second-largest mountain lake in the world behind Lake Titicaca in South America”.

Fixed

- Change “keep” to “keeps”.

Fixed

- Change “In study [5] authors…” to, for example, “In 2021, Rojas-Jimenez and colleagues [5]…”

Fixed

- Delete “A” from “A depth and temperature…”

Fixed

- Delete “the” from “…with the variations in Cyanobacteria”.

Fixed

- In the sentence, “The dominance of Planctomycetes and Chloroflexi in the deepest layers can only be seen in a few lakes of the world”. Is that true? If so, references are required.

We have removed this offer

- Change “in bottom sediments for Issyk-Kul” to “in bottom sediments of Issyk-Kul Lake”.

Fixed

- Change “in comparison with” to “in comparison to”.

Fixed

- Rephrase “In study [10]…”

Fixed

- Delete comma after i.e.

Fixed

- In the sentence “The photosynthetic activity at the bottom of the lake, […], seem to trigger the epitaxial precipitation of the oriented fibers of vaterite”, why these findings are relevant for this study?

We have removed this offer

- Issyk-Kul should always come with the word “Lake”.

Fixed

- Delete “Issyk-Kul is an important center of tourism and recreation in Kyrgyzstan. However,”. Further in the same paragraph, why is the ecosystem of Issyk-Kul Lake fragile? This paragraph also needs to be rephrased.

Fixed

- Paragraph starting with “Of greatest concern is mining and its waste” also needs to be rephrased. Consider including only what is relevant for this study.

- Delete dot in “Issyk-Kul Lake, to study…”

We have removed this offers

PHYSIOGRAPHIC BACKGROUND AND METHODS

- Change “degrees Celsius” to “ºC”.

Fixed

- Change “monomietie”. What is it in English?

Fixed

- Change mg/I to mg/l (with an L and not with an I). It is miligrams/liter.

Fixed

- Figure1: consider changing the sampling points to IK-1, IK-2, IK-3… as in Table1 and in the text.

Fixed

- Change Stotal to Stotal, S(VI) to SO42-, and S (II) to S2+.

Change Stotal to Stotal, — fixed

Change… S(VI) to SO42-, and S (II) to S2+ — in geochemistry, such a designation is accepted: S (II) and S (VI).

https://pubs.acs.org/doi/pdf/10.1021/acscatal.1c01201

- The part where you explain environmental 16S rRNA gene analysis is not well described and text is missing.

fixed

Also, references are missing regarding the preparation of libraries.

Fixed

- Subsections are missing as well, since there is a subsection titled “geomechical modelling”.

Fixed

-What is llnl?

Fixed (Lawrence Livermore National Laboratory https://rdrr.io/cran/phreeqc/man/llnl.dat.html)

RESULTS

- Delete “active” in “active microbial activity”.

Fixed

- In subsection 3.3., why do you make the statement “all of them are organotrophic bacteria”? At that taxonomic level, you cannot conclude that all Proteobacteria or Verrucomicrobiota (for example) are organotrophic. Do you have further data to support that conclusion?

Phrase was removed

- Rephrase “Representatives of the bacterial phylum of the reducing branch of the sulfur cycle Desulfobacterota were found in all samples”.

Corrected

- Change “oxidization” to “oxidation”.

Fixed

- Rephrase paragraph “The second most representative family…”.

Done

- Change “Spirohaetaceae” to “Spirochaetaceae”.

Fixed

- In the paragraph “Green bacteria observed in bottom sediment samples…”, then you refer to phototrophic Acidobacteria, PURPLE sulfur bacteria, etc. Are you saying purple sulfur bacteria are green bacteria?

fixed

- What is a diversity index? How did you measure it?

The diversity indices (Simpson, Shannon, Equitability) were calculated using the PAST 4.06b program. OTUs obtained after 16s rRNA sequencing data in the SILVAngs database ((https://ngs.arb-silva.de/silvangs/) were used to calculate diversity indices.

DISCUSSION

Results need to be well understood to consider the peer-review of the discussion.

We tried to redo the discussion and gave a clearer description of the results

Reviewer 2 Report

1. Too many figures and tables are not giving the main message of the study to audience. Unless absolutely required, moving some of the tables and figures to supporting information is highly recommended. 

2. Conclusion needs better correlation between the microphotographs and energy dispersion spectra, and the microbial community cluster data. 

Author Response

Reply to REW 2

  1. Too many figures and tables are not giving the main message of the study to audience. Unless absolutely required, moving some of the tables and figures to supporting information is highly recommended.

Fixed

  1. Conclusion needs better correlation between the microphotographs and energy dispersion spectra, and the microbial community cluster data.

Fixed